# VCoT: Visual Chain-of-Thought for Continual Learning in Day-Night Object Tracking

## Abstract

Stable tracking in both daytime and nighttime is essential for applying single object tracking to real-world scenarios. Traditional daytime trackers mainly rely on clear appearance features, which leads to significant performance degradation under nighttime conditions. Conversely, nighttime trackers often incorporate low-light enhancement techniques to improve robustness but struggle to maintain comparable accuracy in daytime environments. To address this challenge, we propose a novel framework, termed Visual Chain-of-Thought (VCoT), which reformulates object tracking as a structured reasoning process. VCoT follows a three-stage cognitive path of Observe–Recall&Infer–Memorize: it first observes and extracts the appearance and motion features of the current frame; then retrieves and fuses relevant historical prompts from a memory pool via an attention mechanism to enable context-aware reasoning; and finally employs gradient-based importance evaluation to update the memory by selectively retaining the most valuable knowledge. This design allows the model to integrate real-time observations with historical experiences, while achieving continual learning and effective knowledge transfer across tasks. Extensive experiments on multiple challenging benchmarks demonstrate that VCoT consistently outperforms existing methods under diverse illumination conditions. Codes will be available at https://github.com/Gkk10/VCoT.

## 1 Introduction

Object tracking plays a vital role in applications such as search-and-rescue drones Mishra et al. (2020); Martinez-Alpiste et al. (2021); Abdelnabi & Rabadi (2024), nighttime border patrol Bhanuprakash et al. (2025); Sharma et al. (2021), and urban surveillance Mohanty et al. (2025); Abba et al. (2024); Liu et al. (2021b). These tasks usually require tracking systems to maintain stable and reliable perception across two drastically different lighting environments: daytime and nighttime. For example, in earthquake rescue missions Calamoneri et al. (2022); Papyan et al. (2024), drones need to continuously search suspicious areas across day and night; in nighttime security or border patrol tasks Koslowski & Schulzke (2018); Misbah et al. (2023), the system must still accurately localize targets even under poor appearance visibility. If a system only works under a single lighting condition, its practicality in real-world scenarios will be severely limited. Therefore, developing a unified tracking mechanism with strong generalization across both daytime and nighttime scenes has become a key step toward making visual tracking truly applicable in practice.

Existing object tracking algorithms often perform well under specific lighting conditions such as daytime or nighttime. For example, ProContEXT Lan et al. (2023) achieves precise target localization in daylight scenes with sufficient illumination and clear textures by relying on appearance features. In contrast, DCPT Zhu et al. (2024a) improves tracking robustness in low-light environments by introducing the mechanism of darkness clue prompts. However, these methods are usually designed exclusively for either daytime or nighttime scenarios: daytime trackers are typically effective only under bright conditions, while nighttime trackers are tailored to low-light settings. When such single-condition methods are deployed in real-world applications that require continuous operation across day and night—such as search-and-rescue drones or surveillance systems—their performance may degrade rapidly under unseen lighting conditions. This limitation significantly restricts the reliability and practicality of these systems. These challenges highlight the necessity of design-

ing a tracking mechanism that can maintain stable performance across both daytime and nighttime environments.

From the perspective of human-like cognition, current mainstream tracking methods Zhou et al. (2020); Voigtlaender et al. (2020) face two major limitations. First, most approaches Bertinetto et al. (2016); Li et al. (2019a) focus only on single-frame information and lack the ability to model temporal continuity. As a result, they struggle to reason about motion changes through multi-step inference in the way humans do. Second, when leveraging historical information Danelljan et al. (2019); Bhat et al. (2019), these models cannot selectively retain or flexibly transfer knowledge, which makes it difficult for them to accumulate experience and adapt quickly when the environment changes. In contrast, humans, when facing uncertain situations such as occlusion, blur, or incomplete information, typically observe the motion trend of the target, recall past experiences, infer the potential position, and remember the most critical information. This process illustrates how humans integrate observations with memory, enabling them to accumulate knowledge while improving perception across different scenarios and tasks.

Motivated by the success of the Chain-of-Thought (CoT) Wei et al. (2022) mechanism in large language models for complex reasoning tasks, this paper introduces a Visual Chain-of-Thought (VCoT) framework to enable unified cognitive reasoning across both daytime and nighttime tracking scenarios. We further formulate day–night tracking as a continual learning problem, where the model must adapt between tasks under different illumination conditions while avoiding the loss of previously acquired knowledge. The overall architecture of VCoT is illustrated in Fig. 1. VCoT unfolds along a three-stage cognitive pathway of Observe–Recall&Infer–Memorize: Observe: extract appearance features from the current frame to encode the target's visual state and motion trend, generating observation prompts; Recall-Infer: use the current observation prompt as a query to retrieve and integrate relevant historical knowledge from the prompt pool, producing context-aware reasoning signals that guide Transformer sub-modules for structural modeling and decision making; Memorize: apply gradient-based importance weighting to evaluate newly generated prompts, selectively retaining the most representative knowledge to support accumulation and preservation across tasks.

The main contributions of this work are summarized as follows: **1)** VCoT framework. We are the first to introduce the concept of chain-of-thought reasoning into visual object tracking. By designing a cognitive process of Observe–Recall&Infer–Memorize, our model can maintain effective feature extraction and deliver stable tracking performance across both daytime and nighttime environments. **2)** Prompt-based continual learning. We construct a prompt pool as a memory unit and employ gradient-based importance weighting for selective updating, enabling dynamic adaptation to new tasks while effectively alleviating catastrophic forgetting. **3)** Extensive evaluation. Experiments on multiple daytime and nighttime benchmarks demonstrate that VCoT achieves superior cross-scenario generalization and overall performance compared to state-of-the-art methods.

## 2 RELATED WORKS

### 2.1 OBJECT TRACKING ACROSS DAYTIME AND NIGHTTIME SCENES

Maintaining stable tracking performance under varying illumination remains a long-standing challenge in single object tracking. Most existing methods Held et al. (2016); Bertinetto et al. (2016) rely heavily on appearance-based similarity learning. These approaches typically achieve strong results in daytime scenarios with good lighting and clear textures. For instance, representative trackers such as OSTrack Ye et al. (2022a); Chen et al. (2024) leverage rich visual appearance cues to deliver accurate performance on standard well-lit benchmarks. However, in nighttime or low-light environments, image quality degrades severely, with blurred contours and missing texture information, making appearance-driven trackers struggle to maintain robustness. To address this issue, some studies Li et al. (2019b); Luo et al. (2025) introduce external prompts as complementary signals. For example, DCPT Zhu et al. (2024a) incorporates darkness-related prompts to enhance noise resistance in low-light conditions. Despite these efforts, such methods are often optimized for a single illumination domain. When deployed in real-world applications like inspection drones or surveillance systems that require continuous operation across both daytime and nighttime, their performance typically drops significantly in non-target illumination scenarios.

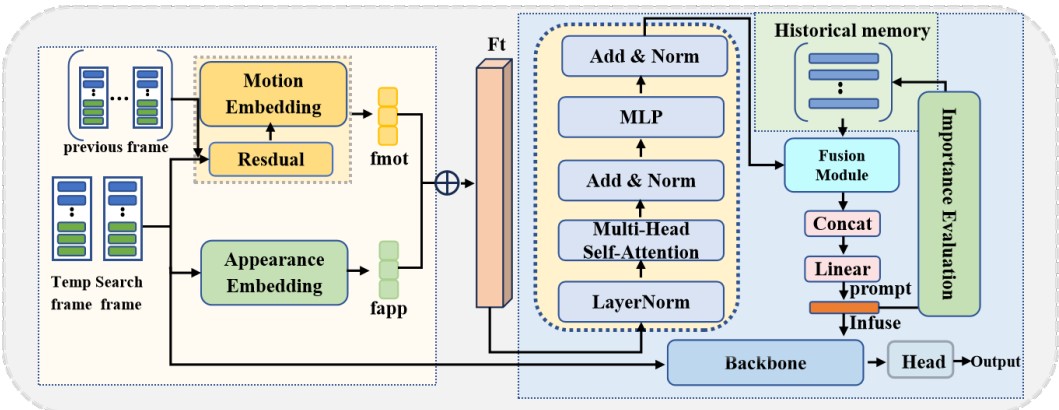

Figure 1: Schematic illustration of the overall VCoT framework. The current frame's appearance features and residual-based motion features are first extracted and fused. The fused features then interact with historical prompts in the memory via a query function, enabling the "recall-infer" process. The generated prompts are injected into the backbone, and finally, based on importance weight estimation, selectively stored in the prompt pool.

## 2.2 CONTINUAL LEARNING IN VISION TASKS

The core problem that continual learning (CL) Rebuffi et al. (2017); Shin et al. (2017) aims to solve is how to enable a model to continuously accumulate and transfer knowledge while adapting to new tasks or environments, with minimal forgetting of previously learned information. In recent years, researchers have explored several strategies for mitigating forgetting in tasks such as classification and detection Li & Hoiem (2017); Shmelkov et al. (2017). These include using regularization techniques to preserve prior knowledge and employing sample replay to reduce forgetting. However, in the field of visual object tracking, related studies remain relatively limited. Existing trackers are often tailored to a single task, and their performance tends to degrade significantly when the environment changes. Introducing continual learning into tracking Liu et al. (2023b); Choi et al. (2022) not only helps alleviate performance degradation when switching between daytime and nighttime tasks but also provides new insights for addressing similar cross-domain challenges in future research.

## 2.3 INSPIRATION FROM PROMPT LEARNING AND CHAIN-OF-THOUGHT

Over the past few years, prompt learning and chain-of-thought (CoT) techniques have achieved remarkable success in natural language processing Wei et al. (2022) and multimodal tasks Liu et al. (2023a). Prompt learning Li & Liang (2021); Liu et al. (2021a) guides models to adapt to different task scenarios by embedding learnable prompt vectors into the current task. Chain-of-thought reasoning Yao et al. (2023), on the other hand, tackles complex problems by decomposing them into multiple steps, enabling models to gradually analyze and derive results in a step-by-step manner. Some prior studies Wang et al. (2022) have applied prompt mechanisms to continual learning, while others Hao et al. (2024) have explored the use of CoT for handling complex scenarios. However, most of these efforts remain limited to static image settings, lacking effective modeling of temporal dynamics and historical experience. Motivated by these insights, this work integrates prompt learning with chain-of-thought reasoning and introduces a Visual Chain-of-Thought (VCoT) framework.

## 3 METHOD

### 3.1 VISUAL PROMPT GENERATOR

This paper introduces the Visual Chain-of-Thought (VCoT) framework, which draws inspiration from human cognitive processes to achieve unified target modeling and reasoning across both daytime and nighttime environments. VCoT formulates tracking as a cognitive cycle of "Observe–Recall &Infer–Memorize". Unlike conventional trackers that rely solely on the current frame,

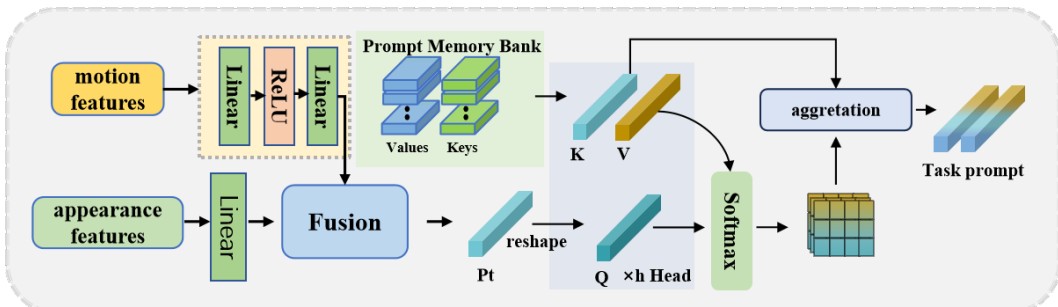

Figure 2: The Prompt Generation and Fusion Process: First, the current task prompts are extracted from the appearance and motion features. Then, they are used as queries to perform attentional fusion with the historical memory prompts (keys/values), resulting in enhanced prompt information.

VCoT actively leverages motion cues and historical experiences to compensate for degraded appearance information in nighttime scenarios, while its dynamic memory update mechanism supports continual learning. This enables the model to maintain stable performance across different illumination domains. the prompt generation and fusion process is illustrated in Fig. 2. Specifically, appearance features are first encoded through a linear transformation, and motion features are extracted by computing residuals between the current frame and the previous k frames. These appearance and motion representations are then concatenated along the sequence dimension and fed into a Transformer encoder, where their relationships are jointly modeled to produce fused prompt vectors. This process can be expressed as:

$$p_t^{\text{app}} = W_{\text{app}} x_t,$$
$$p_t^{\text{mot}} = \text{MLP}(x_t - x_{t-1}),$$
$$\hat{p}_t = \text{TransEnc}([p_t^{\text{app}}, p_t^{\text{mot}}]) \in \mathbb{R}^{1 \times D},$$

where, $x_t$ denotes the feature vector of the current frame, while $x_{t-k}$ represents the feature vector from the previous k frames. The terms $p_t^{\text{app}}$ and $p_t^{\text{mot}}$ correspond to the appearance prompt and the motion prompt, respectively, and $\hat{p}_t$ denotes the fused prompt representation. $D$ is the feature dimension. The generated prompt vectors are stored in a prompt pool, which serves as a memory buffer to maintain previously generated prompts. In this way, when new prompts are created, the model can refer to past information and integrate prior experience into the current task as guidance. This process can be formulated as follows:

$$M(Q, K, V) = \text{Concat}(\text{head}_1, \ldots, \text{head}_h) W^O,$$
$$\text{head}_i = \text{softmax}\left(\frac{QW_i^Q(KW_i^K)^T}{\sqrt{d}}\right) VW_i^V,$$

where $Q$ is $\hat{p}_t$, $K$ and $V$ is the historical memory, and $W_i^Q, W_i^K, W_i^V$ are learnable projection matrices. This process enables the model to aggregate contextual information across time and generate updated prompts. The linear layer projects them into a fixed-length prompt sequence $P_t \in \mathbb{R}^{L \times D}$. The prompt is then expanded along the batch dimension and concatenated to the front of the main input token sequence:

$$X_t' = [P_t^{(D)}; X_t].$$

The prompt length is L = 5 and dimension D = 512, consistent with the backbone. Prompts are injected once at the Transformer input and propagate with feature tokens through all layers. Each frame generates one prompt, with memory capacity M = 50 and smoothing window K = 5 for stable importance estimation.

### 3.2 PROMPT MEMORY AND IMPORTANCE EVALUATION

Traditional object tracking algorithms typically rely on modeling the current target, while overlooking the role of historical experience in assisting the ongoing tracking task. Prior studies Cai et al.

(2024) have shown that accumulating historical knowledge can not only improve recognition performance but also help mitigate catastrophic forgetting. This aligns with the goal of day–night tracking, where the objective is to fully leverage complementary information from both daytime and nighttime while avoiding the loss of nighttime knowledge during training. To this end, we maintain a prompt memory module that stores previously generated prompts. When a new prompt is generated, the system determines whether it should be added to memory. Since storage capacity is limited, it is impossible to retain all prompts, making it essential to establish a mechanism for selecting the most valuable ones. We propose a gradient-guided prompt scoring mechanism to evaluate the contribution of each prompt to the reduction of the loss function during model updates. Unlike traditional similarity-based metrics, our method dynamically prioritizes memory retention based on the verified utility of prompts, thereby improving both knowledge consolidation and evolution. The core idea is intuitive: if a prompt has a stronger influence on the loss reduction during training, it is more valuable to retain. Let $P_t \in \mathbb{R}^{L \times D}$ denote the prompt sequence at time step $t$. During backpropagation, we compute the gradient $\nabla_{P_t}\mathcal{L}$ with respect to each token in the sequence. The sensitivity score is obtained by averaging the $\ell_2$-norm of the gradients across all tokens:

$$g_t = \frac{1}{L}\sum_{l=1}^{L}\|\nabla_{p_{t,l}}\mathcal{L}\|_2,$$

which reflects the overall sensitivity of the prompt to parameter updates in a single optimization step. To suppress noise fluctuations and ensure stable scoring, we apply a sliding window of size $K$ to smooth the sensitivity values. The final importance score is then defined as the average over the most recent $K$ steps:

$$s_t = \frac{1}{K}\sum_{k=1}^{K}g_t^{(k)}.$$

After computing $s_t$, each prompt is assigned an importance score and stored in the prompt memory. To prevent unbounded growth, the memory retains only the top-$M$ prompts with the highest importance scores at any given time.

### 3.3 CONTINUAL LEARNING MODELING

For a tracker to adapt robustly across day and night conditions, it must maintain stable perception under varying illumination. Addressing the plasticity–stability dilemma in continual learning is therefore critical. If the model relies solely on the features of the current task without retaining past knowledge, new training will inevitably overwrite previous representations, leading to severe forgetting. To alleviate this, we introduce a memory mechanism that supports selective storage and updating of prompts across tasks. Prompts serve both as contextual cues for the current task and as transferable knowledge units that accumulate experience over time. This allows the model to continually leverage prior knowledge while adapting to new environments, thus achieving cross-task consistency and stability. Specifically, when encountering new tasks, the model evaluates the importance of newly generated prompts and updates the memory pool accordingly. The retained prompts can then be recalled and fused with current observations, enabling knowledge transfer between day and night domains. Formally, let $P_t$ denote the prompt at time $t$ and $s(P_t)$ its gradient-based importance score. The memory update rule is:

$$M_{t+1} = \text{Top}_M\big(M_t \cup \{(P_t, s(P_t))\}\big),$$

where $\text{Top}_M$ selects the top-$M$ prompts with the highest importance scores.

At inference, the current observation $o_t$ (encoded from external inputs) is used as the query, while prompts in the memory serve as keys and values. A multi-head attention module retrieves and integrates the most relevant historical prompts with the current observation:

$$r_t = \text{MHA}(Q = o_t,\ K = P_{\text{hist}},\ V = P_{\text{hist}}),$$

where $P_{\text{hist}}$ denotes the set of prompts stored in memory. In this design, gradient sensitivity governs memory writing and updating, while attention drives memory retrieval and fusion. This complementary mechanism allows the tracker to accumulate knowledge incrementally while maintaining stability, ensuring robust adaptation across day and night tasks.

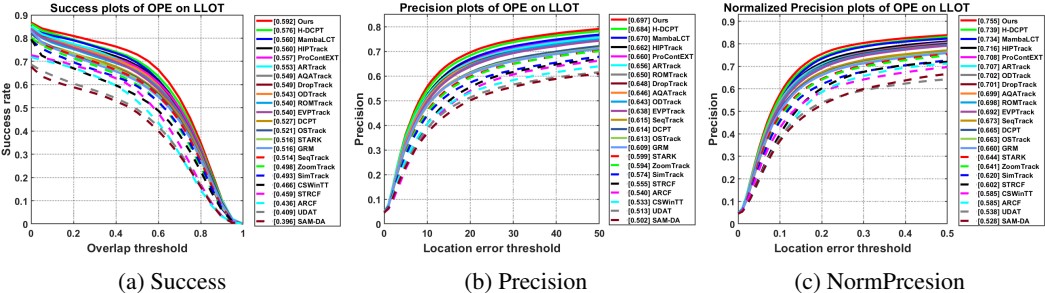

(a) Success        (b) Precision        (c) NormPrcesion

Figure 3: Overall performance of VCoT and other SOTA trackers (DropTrack Durve et al. (2022), ROMTrack Cai et al. (2023), SeqTrack Chen et al. (2023), OSTrack Ye et al. (2022a), GRM Gao et al. (2023), STARK Yan et al. (2021), ZoomTrack Kou et al. (2023), SimTrack Chen et al. (2022), STRCF Li et al. (2018), CSWinTT Song et al. (2022), ARCF Huang et al. (2019b), UDAT Ye et al. (2022c) ) on LLOT. View enlarged image for clarity.

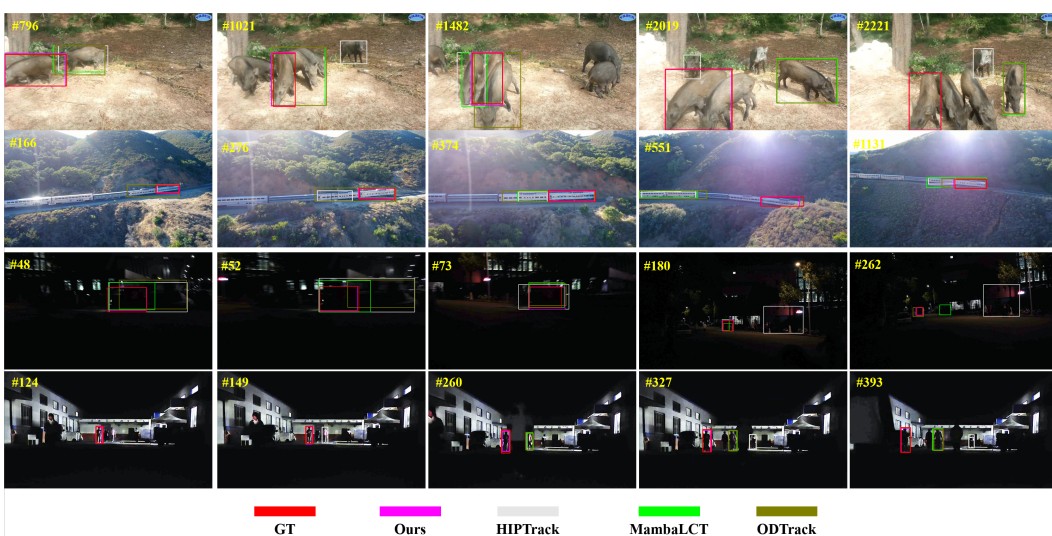

Figure 4: Visual comparison between our tracker and three SOTA methods on LaSOT and LLOT benchmarks.

## 4 EXPERIMENTS

We treat daytime and nighttime tracking as two sequential tasks in a continual learning setting. For training, we construct task-specific datasets under two illumination conditions. The daytime task integrates LaSOT Fan et al. (2019), GOT-10k Huang et al. (2019a), COCO Lin et al. (2014), and TrackingNet Muller et al. (2018) to cover diverse daytime scenarios, while the nighttime task leverages BDD100K-Night Yu et al. (2020) and SHIFT-Night Sun et al. (2022), which contain low-light conditions. For evaluation, DTB70 Li & Yeung (2017), VisDrone2018 Zhu et al. (2018), UAVDT Du et al. (2018), and OTB100 Wu et al. (2013) are used as benchmarks for daytime tasks, whereas UAVDark135 Li et al. (2022), NAT2024-1 Fu et al. (2024a), DarkTrack2021 Ye et al. (2022b), and LLOT Zhong et al. (2025) are employed for nighttime tasks. To further validate the generalization ability of our method, we additionally conduct large-scale experiments on GOT-10k Huang et al. (2019a) and TrackingNet Muller et al. (2018). Performance is evaluated following standard single-object tracking protocols, using Area Under the Curve (AUC), Precision (P), and Normalized Precision ($P_{norm}$) as the main metrics. To ensure fair comparisons, strong baselines such as ODTrack, HIPTrack, H-DCPT and AQATrack are retrained on the same datasets using their official default settings before testing on nighttime benchmarks.

Table 1: Comparison of model parameters(M), MACs(G), and Spees(fps).

| Tracker | Params | MACs | Speed | Device |
|---------|--------|------|-------|--------|
| VCoT(Ours) | 95M | 23G | 30fps | Titan X |
| HIPTrack | 120M | 66G | 25fps | Titan X |
| ODTrack | 92M | 73G | 16fps | Titan X |

## 4.1 IMPLEMENTATION DETAILS

We adopt HiViT-Base as the backbone network, with input sizes of $224 \times 224$ for the search region and $112 \times 112$ for the template region. The model is trained using the AdamW optimizer Loshchilov & Hutter (2017) for 300 epochs with a batch size of 16. The initial learning rate is set to $1 \times 10^{-4}$ and the weight decay to $1 \times 10^{-4}$. Each epoch contains approximately 60,000 image pairs. To stabilize training, the learning rate is reduced to $1 \times 10^{-5}$ after 250 epochs. All experiments are conducted on a workstation equipped with an Intel i9-10850K CPU, 16 GB of memory, and an NVIDIA Titan X GPU. We report the model parameters, computational complexity (MACs), and inference speed alongside other trackers in Table 1. '

Table 2: Comparison of tracking performance on nighttime datasets. The top three results are highlighted in red, blue, and green.

| Method | Source | UAVDark135 AUC | P | $P_{\text{Norm}}$ | NAT2024-1 AUC | P | $P_{\text{Norm}}$ | DarkTrack2021 AUC | P | $P_{\text{Norm}}$ |
|--------|--------|------|------|------|------|------|------|------|------|------|
| VCoT (Ours) | - | 68.6 | 82.6 | 83.9 | 73.3 | 94.5 | 90.7 | 62.6 | 74.2 | 75.0 |
| MambaLCT Li et al. (2025) | AAAI2025 | 64.4 | 77.9 | 80.6 | 60.8 | 88.9 | 85.8 | 61.5 | 74.4 | 75.1 |
| H-DCPTZhong et al. (2025) | T-IV2025 | 64.7 | 77.3 | 77.9 | 71.0 | 90.8 | 86.2 | 62.0 | 74.6 | 74.2 |
| ARTrackWei et al. (2023) | CVPR2023 | 65.5 | 77.8 | 79.3 | 69.9 | 90.9 | 84.3 | 61.2 | 72.5 | 72.0 |
| ODTrack Zheng et al. (2024) | AAAI2024 | 63.2 | 77.8 | 78.1 | 69.1 | 89.6 | 85.5 | 60.5 | 72.2 | 72.3 |
| MCITrackKang et al. (2025) | AAAI2025 | 58.4 | 68.7 | 70.3 | 65.2 | 81.8 | 78.2 | 54.5 | 64.8 | 65.4 |
| HIPTrack Cai et al. (2024) | CVPR2024 | 59.7 | 72.0 | 70.0 | 69.4 | 88.4 | 84.1 | 57.1 | 68.5 | 68.7 |
| EVPTtrack Shi et al. (2024) | AAAI2024 | 58.1 | 69.2 | 70.5 | 65.0 | 83.7 | 78.1 | 53.7 | 64.8 | 64.7 |
| AQATrack Xie et al. (2024) | CVPR2024 | 58.2 | 69.2 | 70.7 | 64.2 | 82.1 | 77.1 | 55.0 | 66.1 | 66.8 |
| SAM-DA Fu et al. (2024b) | ICARM24 | 47.6 | 60.4 | 59.4 | 53.4 | 75.3 | 64.9 | 44.7 | 55.5 | 54.6 |
| DCPT Zhu et al. (2024b) | ICRA2024 | 56.7 | 69.2 | 69.8 | 62.1 | 80.9 | 75.4 | 54.0 | 66.7 | 64.6 |
| AVTrack Li et al. (2024) | ICML2024 | 47.6 | 58.6 | 59.2 | 56.7 | 68.2 | 75.3 | 46.1 | 55.1 | 54.9 |
| LiteTrack Wei et al. (2024) | CVPR2024 | 53.9 | 63.6 | 65.9 | 61.8 | 79.7 | 74.1 | 52.8 | 63.5 | 62.8 |

Table 3: Comparison of tracking performance on daytime datasets. The top three results are highlighted in red, blue, and green.

| Method | Source | DTB70 AUC | P | $P_{\text{Norm}}$ | VisDrone2018 AUC | P | $P_{\text{Norm}}$ | UAVDT AUC | P | $P_{\text{Norm}}$ |
|--------|--------|------|------|------|------|------|------|------|------|------|
| VCoT (Ours) | - | 70.1 | 90.7 | 85.3 | 70.6 | 90.0 | 86.8 | 67.2 | 88.8 | 77.6 |
| MambaLCT Li et al. (2025) | AAAI2025 | 68.7 | 88.3 | 84.0 | 65.4 | 88.1 | 84.3 | 63.6 | 84.4 | 75.8 |
| H-DCPT Zhong et al. (2025) | T-IV2025 | 68.6 | 77.3 | 77.9 | 68.5 | 88.5 | 85.6 | 61.8 | 81.6 | 74.1 |
| ARTrack Wei et al. (2023) | CVPR2023 | 66.4 | 87.5 | 81.1 | 65.9 | 87.4 | 82.9 | 63.8 | 85.1 | 74.0 |
| ODTrack Zheng et al. (2024) | AAAI2024 | 70.0 | 90.0 | 86.1 | 64.7 | 85.6 | 83.1 | 63.8 | 85.8 | 74.7 |
| MCITrack Kang et al. (2025) | AAAI2025 | 68.1 | 87.4 | 82.0 | 66.7 | 84.3 | 81.2 | 62.1 | 81.5 | 73.3 |
| HIPTrack Cai et al. (2024) | CVPR2024 | 68.6 | 81.2 | 76.2 | 67.1 | 86.7 | 83.9 | 60.9 | 81.2 | 76.2 |
| EVPTtrack Shi et al. (2024) | AAAI2024 | 66.6 | 86.7 | 81.7 | 66.6 | 87.0 | 82.6 | 60.2 | 80.0 | 71.3 |
| AQATrack Xie et al. (2024) | CVPR2024 | 66.1 | 86.3 | 80.7 | 66.9 | 87.2 | 89.8 | 63.7 | 84.7 | 75.9 |
| SAM-DA Fu et al. (2024b) | ICARM24 | 63.0 | 82.2 | 76.3 | 53.1 | 71.4 | 67.0 | 61.3 | 82.6 | 73.3 |
| DCPT Zhu et al. (2024b) | ICRA2024 | 64.6 | 83.7 | 77.6 | 64.2 | 83.1 | 79.7 | 56.9 | 76.8 | 66.0 |
| AVTrack Li et al. (2024) | ICML2024 | 65.0 | 84.3 | 80.0 | 64.2 | 84.8 | 80.3 | 58.7 | 82.1 | 68.6 |
| LiteTrack Wei et al. (2024) | CVPR2024 | 64.7 | 83.5 | 77.6 | 61.8 | 79.8 | 75.7 | 62.1 | 84.3 | 71.6 |

Table 4: Performance comparison on Track-ingNet dataset.

| Method | AUC | $P$ | $P_{Norm}$ |
|--------|-----|-----|-----------|
| VCoT (Ours) | 85.02 | 84.60 | 89.52 |
| ManBaLCT | 84.30 | 83.90 | 89.20 |
| HIPTrack | 84.50 | 83.80 | 89.10 |
| LoRAT | 83.50 | 82.10 | 87.90 |
| ARTrackV2 | 84.90 | 84.50 | 89.30 |
| EVPTtrack | 83.50 | - | 88.30 |
| AQATrack | 83.80 | 83.10 | 88.60 |
| LiteTrack | 80.80 | 78.20 | 85.70 |

Table 5: Performance comparison on GOT-10K dataset.

| Method | AO | $SR_{0.5}$ | $SR_{0.75}$ |
|--------|-----|-----|-----------|
| VCoT (Ours) | 77.10 | 87.01 | 76.70 |
| MambaLCT | 74.80 | 85.40 | 72.10 |
| HIPTrack | 77.40 | 88.70 | 74.50 |
| LoRAT | 72.10 | 81.80 | 70.70 |
| ARTrackV2 | 75.90 | 85.40 | 72.70 |
| EVPTrack | 73.30 | 83.60 | 70.70 |
| AQATrack | 73.80 | 83.20 | 73.10 |
| LiteTrack | 68.70 | 78.20 | 64.20 |

Table 6: Performance comparison on LaSOT dataset

| Method | AUC | P | Norm P |
|--------|-----|-----|-----------|
| VCoT(Ours) | 72.24 | 78.90 | 82.32 |
| MambaLCT | 71.80 | 79.46 | 83.09 |
| HIPTrack | 72.70 | 79.50 | 82.90 |
| LoRAT | 71.70 | 77.30 | 80.90 |
| ARTrackV2 | 71.60 | 80.60 | 83.20 |
| EVPTrack | 70.40 | 77.20 | 80.90 |
| AQATrack | 71.40 | 81.90 | 78.60 |
| LiteTrack | 64.60 | 68.90 | 73.90 |

Table 7: Ablation study under intra-sequence illumination changes on LLOT.

| Method | AUC |
|--------|-----|
| Baseline | 0.539 |
| Baseline + Prompt | 0.542 |
| Baseline + Memory | 0.561 |
| Baseline +Prompt +Memory | **0.572** |

## 4.2 COMPARISON WITH STATE-OF-THE-ART METHODS

**Quantitative Comparison** On nighttime benchmarks, our method demonstrates clear performance advantages. As shown in Table 2 and Fig. 3, VCoT achieves 68.61% AUC and 82.69% P on UAVDark135, outperforming all competing state-of-the-art methods. On Nat2024-1, VCoT ranks first with 73.33% AUC, and 94.57% P, highlighting its robustness to appearance degradation under low-light conditions. Furthermore, on two particularly challenging datasets, DarkTrack2021 and LLOT, our method consistently maintains leading results across all three metrics, validating the stability and generalization ability of VCoT under diverse illumination scenarios.

On daytime benchmarks, VCoT likewise surpasses existing SOTA trackers. As presented in Table 3, on DTB70, our method achieves 70.13% AUC and 90.75% P. On OTB100, it further obtains 71.76% AUC, achieving the best overall tracking performance. On UAVDT, VCoT delivers significant improvements with 67.25% AUC and 88.81% P, reflecting strong adaptability to complex real-world conditions. On VisDrone2018, it reaches 70.64% AUC, 90.02% P, and 86.84% $P_{Norm}$, outperforming all competitors and confirming its effectiveness under scale variation and motion blur. VCoT demonstrates remarkable performance in the OTB100 radar chart (Fig. 5), exhibiting outstanding advantages in handling core challenges such as partial occlusion and motion blur.

To further verify generalization and scalability, we conduct experiments on two large-scale datasets, LaSOT, GOT-10k and TrackingNet. As reported in Table 4 and 5, 9, VCoT consistently achieves strong results. On GOT-10k, it achieves higher AO and SR compared to MambaLCT and LoRAT Lin et al. (2024). On TrackingNet, it surpasses HIPTrack and ARTrackV2 Bai et al. (2024) across AUC, P, and $P_{Norm}$, demonstrating robust generalization in large-scale real-world scenarios.

**Qualitative Comparison** To further validate the effectiveness of the proposed method under varying illumination conditions, we conduct qualitative comparisons with representative state-of-the-art trackers across diverse day and night scenarios, as illustrated in Fig. 4. In daytime scenes (first and second rows), mainstream methods such as HIPTrack, MambaLCT, and ODTrack perform reasonably well when the target appearance is clear. However, they often suffer from boundary shifts or target loss under strong illumination or background distractions. In contrast, our method consistently maintains accurate boundary alignment, leading to more stable and precise tracking. In nighttime scenarios (third and fourth rows), low illumination and noise pose significant challenges,

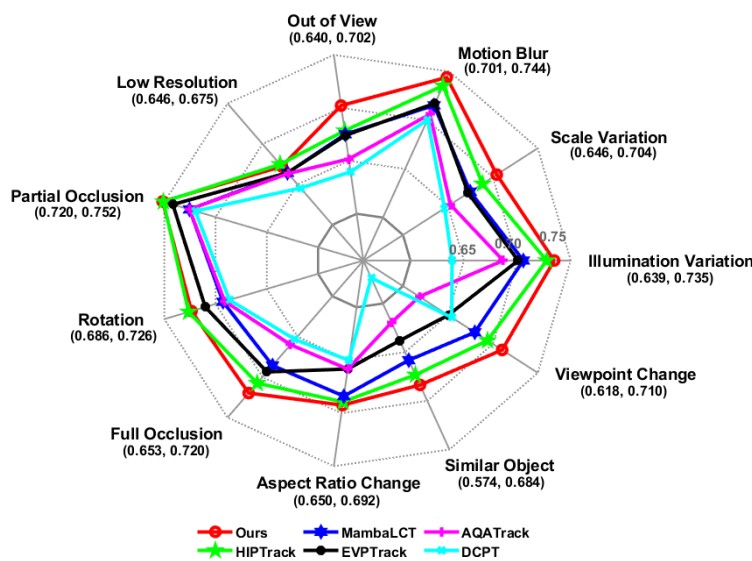

Figure 5: Performance Comparison Across OTB100 Challenge Attributes.

where existing methods commonly exhibit bounding box drift or incorrect matches. For example, HIPTrack tends to drift under weak illumination, MambaLCT struggles when the contrast between target and background is low, and ODTrack fails to capture the target under complex lighting conditions. By leveraging the proposed Visual Chain-of-Thought reasoning mechanism to effectively integrate historical memory, our method is able to robustly localize the target even in extremely poor illumination and heavy occlusion cases.

Table 8: Stepwise Ablation Results of VCoT Components on LLOT and DTB70. O, R, and M correspond to the Observe, Recall-Infer, and Memorize stages in our VCoT framework.

| Method | LLOT | | DTB70 | |
|---|---|---|---|---|
| | Succ. | Prec. | Succ. | Prec. |
| Baseline | 55.88 | 62.90 | 67.85 | 87.39 |
| Baseline+O | 57.52 | 63.85 | 68.72 | 89.02 |
| Baseline+O+R | 58.47 | 64.91 | 69.20 | 88.90 |
| Baselinee+O+R+M | **59.38** | **65.42** | **70.13** | **90.75** |

Table 9: Ablation study on continual learning ability in LaSOT benchmark.

| Train Data | Method | Succ. | Prec. |
|---|---|---|---|
| Daytime only | Baseline | 70.99 | 77.04 |
| Daytime+Nighttime | Baseline | 67.49 | 72.71 |
| Daytime +Nighttime | Replay | 69.45 | 75.13 |
| Daytime +Nighttime | VCoT | **72.74** | **78.87** |

Table 10: Ablation study on the role of the historical prompt pool on DTB70 dataset.

| Method | Succ. | Prec. |
|---|---|---|
| Baseline | 67.85 | 87.39 |
| Baseline+Prompt | 68.15 | 87.24 |
| VCoT | **70.13** | **90.75** |

## 4.3 ABLATION STUDY

**Stepwise Ablation of VCoT Components.** As shown in Table 8. We first evaluate the independent contributions of the three stages: Observe (O), Recall-Infer (R), and Memorize (M) on the LLOT and DTB70 datasets. The baseline model (B) achieves only 55.88%/62.9% success and precision on LLOT. Adding the Observe module (B+O), then the Recall-Infer stage (B+O+R), and finally the full three-stage pipeline (B+O+R+M), the performance improves step by step, reaching

59.38%/65.42% on LLOT. On DTB70, the complete model achieves 70.13% success and 90.75% precision, surpassing the baseline by 2.28% and 3.36%, respectively. Each stage proves useful, and Memorize especially strengthens stability across day and night.

**Continual Learning Capability.** As presented in Table 9. To evaluate the model's continual learning ability under day-to-night task switching, we conduct staged training experiments on LaSOT. When trained only on daytime data, the model achieves 70.99%/77.04% in success and precision. However, when subsequently trained on nighttime data without an effective cross-task memory mechanism, the performance drops to 67.49%/72.71%, indicating clear forgetting. In contrast, with our proposed continual learning scheme, the model is able to retain both daytime and nighttime knowledge, achieving 72.74%/78.87%. We also compare with the classic replay-based continual learning method (third row in Table 9). Although experience replay mitigates catastrophic forgetting to some extent, it only provides passive, sample-level compensation and struggles to maintain stability under abrupt illumination changes or severe appearance degradation. Consequently, its overall performance remains limited, showing a clear gap with our proposed mechanism of prompt-guided inference and selective memory updating.

**Effect of the Historical Prompt Pool.** As illustrated in Table 10. We further evaluate the impact of the historical prompt pool on model performance. As shown in the experiments, when relying only on the current frame's appearance and motion prompts (B), the model achieves 67.85%/87.39% in success and precision on DTB70. Simply introducing prompts without retaining historical memory (B+Prompt) yields little improvement. In contrast, the full method, which leverages the prompt pool to accumulate and selectively retain historical information, improves performance to 70.13%/90.75%. This result underscores the prompt pool's role in knowledge selection and transfer.

**Ablation of prompt and memory modules under illumination changes.** To verify the stability of VCoT under continuous illumination changes, we selected segments with pronounced lighting variations within the same sequence from the LLOT dataset for testing. As shown in Table 7 the baseline model achieves an AUC of 0.539; adding motion prompts marginally improves it to 0.542, indicating that dynamic cues provide some compensation when appearance degrades due to illumination changes; incorporating the memory mechanism further boosts performance to 0.561, demonstrating that historical information offers greater robustness to lighting variations; combining both components reaches the highest AUC of 0.572.

## 5 CONCLUSION

This study is the first to introduce continual learning and the Chain-of-Thought (CoT) reasoning mechanism into visual object tracking. We propose a novel framework, termed Visual Chain-of-Thought (VCoT), which models the human cognitive process of "Observe–Recall&Infer–Memorize." By effectively integrating real-time observations with historical experience, VCoT maintains stable and accurate tracking performance across both daytime and nighttime scenarios. Furthermore, we design a continual learning mechanism that employs gradient-guided importance evaluation to update and retain critical historical information, enabling the model to adapt to new tasks while alleviating the common problem of catastrophic forgetting. Extensive experiments demonstrate that VCoT consistently outperforms current state-of-the-art methods in overall performance.

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
