# OpenReview forum: "VCoT: Visual Chain-of-Thought for Continual Learning in Day-Night Object Tracking"
_ICLR.cc/2026/Conference — Submitted to ICLR 2026_

### Official Review · Reviewer_RZyp · 2025-10-22

**Soundness:** 3
**Presentation:** 2
**Contribution:** 2
**Rating:** 2
**Confidence:** 4

**Summary:**

This paper introduces a novel object tracking framework termed the Visual Chain-of-Thought (VCoT), which is designed to address the challenge conventional trackers encounter in simultaneously adapting to the two disparate illumination conditions of daytime and nighttime.

The core concept of this framework is to emulate the human cognitive process, structuring it as a three-stage "Observe-Recall-Infer-Memorize" cycle:

Observation: Extracting the appearance and motion features of the current frame.

Recall-Inference: Fusing the current observation with historical experiences stored in a "memory pool" to perform inference.

Memorization: Employing a continual learning mechanism to evaluate the importance of new experiences (based on gradients) and selectively storing the most valuable information into the "memory pool".

**Strengths:**

The application scenarios and the inherent difficulties of the task are accurately demonstrated through illustrative examples.

An analysis of various challenging factors is conducted; furthermore, the qualitative comparison analyzes the distinct characteristics of VCoT and other trackers.

Interpreting the tracking problem from the perspective of Chain-of-Thought (CoT) presents a novel and insightful approach.

**Weaknesses:**

Lack of Novelty: While the paper interprets the utilization of temporal information from the perspective of Chain-of-Thought (CoT), it fundamentally amounts to conventional temporal information processing. Furthermore, the contributions exhibit significant overlap with existing works such as ARTrack, ARTrackv2, and MixViT, appearing to be merely a combination of established methods with minor modifications.

Restricted Application Scenarios: The method's applicability is limited; ultimately, it solely addresses the challenge of illumination variations.

Incomplete SOTA (State-of-the-Art) Comparison: The paper does not include comparisons against the latest trackers (MCITrack, etc). Additionally, evaluations on more general-purpose datasets, such as LaSOT, are absent.

ARTrack: Wei, Xing, et al. "Autoregressive visual tracking." Proceedings of the IEEE/CVF conference on computer vision and pattern recognition. 2023.
ARTrackV2: Bai, Yifan, et al. "Artrackv2: Prompting autoregressive tracker where to look and how to describe." Proceedings of the IEEE/CVF conference on computer vision and pattern recognition. 2024.
MixViT: Cui, Yutao, et al. "Mixformer: End-to-end tracking with iterative mixed attention." Proceedings of the IEEE/CVF conference on computer vision and pattern recognition. 2022.
MCITrack: Kang, Ben, et al. "Exploring enhanced contextual information for video-level object tracking." Proceedings of the AAAI Conference on Artificial Intelligence. Vol. 39. No. 4. 2025.

**Questions:**

Does the proposed methodology incorporate any unique design elements specifically targeting the challenge of day-night transitions?

Furthermore, is its applicability restricted to this context, or can the approach be generalized to other challenges or scenarios?

---

> ### Author Response · Authors · 2025-11-20
>
> 1.**Response to comment**: (Lack of Novelty: While the paper interprets .....methods with minor modifications.)
>
> **Response**: We thank the reviewer for the concerns regarding the novelty of our approach. We clarify below the substantive differences between VCoT and methods such as ARTrack and MixViT.
>
> **(1)Difference from ARTrack / ARTrackV2**
> ARTrack adopts an autoregressive modeling mechanism, focusing on sequential prediction of target states. In contrast, VCoT introduces a cognition-inspired structured reasoning loop (Observe–Recall&Infer–Memorize) designed to establish a content-dependent, cross-frame reasoning process. Through dynamic prompt generation and selective retrieval of historical information, VCoT explicitly models long-range temporal dependencies. This mechanism is fundamentally different from the autoregressive paradigm used in ARTrack.
>
> **(2)Difference from MixViT**
> MixViT primarily aims to improve template–search feature fusion through mixed attention. VCoT, however, incorporates an independent external memory module and employs a gradient-based importance strategy for selective update and retrieval of prompt units, enabling long-term temporal knowledge retention and recall. This importance-driven prompt management mechanism goes beyond instantaneous feature fusion and reflects a different design principle.
>
> **(3)Core novelty of VCoT**
> The key contribution of VCoT lies in proposing a structured visual reasoning loop that integrates dynamic prompt generation, cross-frame experience retrieval, and gradient-driven memory maintenance into a unified framework. This design provides temporally consistent reasoning and continual learning capabilities, yielding clear advantages in non-stationary scenarios such as day–night transitions. To our knowledge, such a structured reasoning mechanism has not been explored in prior tracking research.
>
>
>
> 2.**Response to comment**: (Restricted Application Scenarios: The method's applicability is limited; ultimately, it solely addresses the challenge of illumination variations.)
>
> Response: Thank you for raising this point. We would like to clarify that the core objective of this work is not solely to address illumination variation, but rather to tackle a more fundamental and universal challenge induced by drastic environmental changes: catastrophic forgetting in visual object tracking. Illumination variation serves as a typical and highly challenging testing scenario we selected to validate this core capability. The fundamental contribution of the VCoT framework lies in its structured reasoning cycle of "Observe-Recall-Memorize" and its gradient-based memory update mechanism, which collectively address the problem of catastrophic forgetting—enabling the model to adapt to new tasks (e.g., nighttime tracking) without losing knowledge of previously learned tasks (e.g., daytime tracking). Therefore, our method simultaneously targets two closely interconnected challenges:
>
> **(1) The immediate challenge of illumination variation**: compensating for degraded appearance information at night through motion prompts and the recall of historical experience.
>
> **(2) The continual challenge of cross-domain learning**: ensuring that the model retains its ability to perform daytime tracking even after learning to track at night, via a selective memory mechanism.
>
> **3.Response to comment**: (Incomplete SOTA (State-of-the-Art) Comparison: The paper does not include comparisons against the latest trackers (MCITrack, etc). Additionally, evaluations on more general-purpose datasets, such as LaSOT, are absent.)
>
> Response: We have added the latest comparison methods—MCITrack, H-DCPT, and ARTrack—along with their evaluation results on the LaSOT dataset in the revised version. Please refer to Reviewer i22L for the experimental results.
>
>
> 4.**Response to comment**: (Does the proposed methodology incorporate any unique design elements specifically targeting the challenge of day-night transitions? )
>
> **Response**: The proposed method explicitly includes design elements tailored for day–night transitions. The motivation of VCoT is that illumination changes lead to severe degradation of appearance cues, causing standard trackers to drift. To address this, we introduce (1) motion-residual prompts, which provide a stable cue when appearance becomes unreliable at night; (2) a memory-based recall and inference mechanism, which retrieves illumination-consistent historical representations to compensate for appearance collapse; and (3) gradient-based memory maintenance, which prioritizes experience from illumination-transition frames during training. These components form a reasoning loop that enables VCoT to adaptively shift its reliance from appearance to motion and historical cues when transitioning from day to night. Together, they constitute a dedicated mechanism specifically designed to handle illumination transitions, rather than a generic prompt/memory framework.

---

### Official Review · Reviewer_5Xte · 2025-10-26

**Soundness:** 3
**Presentation:** 3
**Contribution:** 3
**Rating:** 6
**Confidence:** 3

**Summary:**

The paper proposes VCoT (Visual Chain-of-Thought) for single-object tracking that must work across day/night. The core idea is to reformulate tracking as a reasoning loop (Observe, Recall–Infer, Memorize), where the tracker (i) builds appearance and residual-motion “observation prompts” from the current frame, (ii) retrieves and fuses relevant historical prompts from a memory pool via attention, and (iii) updates that memory using a gradient-based importance score so only useful prompts are retained. Prompts are prepended to backbone tokens and used to guide the transformer. Trained with day/night as sequential tasks (continual learning framing), VCoT reports SOTA or competitive results on both night (UAVDark135, NAT2024-1, DarkTrack2021, LLOT) and day (DTB70, VisDrone2018, UAVDT, OTB100) benchmarks, plus large-scale GOT-10k and TrackingNet; ablations attribute gains to each stage and to the memory pool mitigating forgetting.

**Strengths:**

(1) A prompt-centric tracking framework that explicitly retrieves from historical context rather than relying only on the current frame, with a clear mathematical derivation of prompt creation and injection.

(2) A gradient-guided memory writing rule (top-M by averaged per-token gradient norms with K-step smoothing) that is simple and effective.

(3) A promising framing of day to night training and evidence that naive joint training causes forgetting

(4) Good experimental results

**Weaknesses:**

(1) The paper says prompts are formed from linear-projected appearance and residual-motion features, encoded via a transformer, then projected to a fixed-length tensor and concatenated to tokens. But values of L/D, which layers get prompts, and how many prompts per frame / per level are not specified very well. Similarly, the memory capacity and smoothing/window are not well-stated in the main text. This limits reproducibility and makes Figures 1–2 feel high-level rather than operational.

(2) Although framed as Chain-of-Thought, the method is attention over a prompt memory plus gradient-based selection, which is kind of close to current prompt/memory trackers. The paper claims to be “first to introduce CoT in tracking,” but the novelty seems incremental relative to prompt-based retrieval + memory. What is the new reasoning capability here beyond (i) motion-residual prompts and (ii) memory-augmented attention?

(3) The paper argues that recalling history & using motion prompts compensates for appearance loss at night, but it seems that there is no targeted analysis that directly shows illumination-invariance gains (e.g., day to night within the same scene, or controlled photometric corruptions). The “continual learning” ablation uses LaSOT (day), not a day to night paired scenario; it shows forgetting mitigation, not illumination robustness.

(4) Baselines are “retrained on the same datasets,” but several are not designed for night or for CL; it’s unclear whether night-specialized or prompt-memory baselines are tuned equivalently

Other minor weaknesses:
(5) The abstract calls Observe–Recall–Infer–Memorize a “three-stage” path, but lists four verbs. Though the authors mentioned that Recall-Infer is a single stage, this is quite confusing to readers.

(6) Tables label the method “UniTrack (Ours)” rather than VCoT, which is kind of confusing

**Questions:**

(1) What are L and D exactly (e.g., what do they look like)? Which backbone stages receive prompts? How many prompts per frame and how are positions encoded?

(2) What are the shapes of M (capacity) and K (smoothing window) exactly? What is the write policy/frequency? Is there class/scene stratification?

(3) Is memory updated online at test? If so, how is the importance approximated without gradients? If not, maybe scope “continual learning” to training-time only.

(4) It is interesting to see experiments with controlled day to night transformations (brightness/contrast/noise) on the same sequences, reporting with/without memory and motion prompts

(5) It is also interesting to see FPS/FLOPs/params comparison.

---

> ### Author Response · Authors · 2025-11-20
>
> 1.**Response to comment**: (The paper lacks in major contribution and motivation of research.)
>
> **Response**: We sincerely thank the reviewer for the careful reading and constructive feedback. We provide detailed clarifications to the main concerns as follows:
>
> (1) **Clarification of Prompt Configuration and Memory Architecture**
>
> In the revised paper, we provide full implementation details as requested. The prompt length is set to L=5 and the feature dimension is D=512, matching the backbone feature dimension. Prompts are injected once at the Transformer input stage and propagate jointly with feature tokens throughout the backbone. One prompt is generated per frame. The memory capacity is M=50, and a smoothing window of K=5 is used to stabilize gradient-based importance estimation across frames. Standard learnable positional embeddings are used and concatenated before feeding the tokens into the backbone. These details will be clearly described in Lines 208–211 on page 4 in the revised manuscript to ensure complete reproducibility.
>
> (2)**Explanation of the Reasoning Behavior Beyond Motion Prompts and Memory Attention**
>
> The contribution of VCoT lies in its structured, cognition-inspired reasoning loop, which integrates all components into a coherent “Observe–Recall&Infer–Memorize” cycle. This is not a simple combination of modules, but a system-level multi-step reasoning mechanism: the model actively analyzes the current target state (appearance + motion), uses it to retrieve relevant historical context from memory for reasoning, and then updates the knowledge base via gradient-based importance estimation. This closed-loop reasoning enables the tracker to dynamically shift its reliance across cues when facing severe day–night changes (e.g., relying less on degraded appearance at night and more on motion trends + retrieved experience), thereby providing adaptive compensation for missing or unreliable information.
>
> (3)**Additional Experiments on Illumination Robustness.**
>
> Following the reviewer’s suggestion, we added controlled experiments to evaluate robustness under illumination changes. As shown in Table 7 (page 8), we select sequences from the LLOT dataset that contain clear intra-sequence illumination transitions (e.g., bright→dark or dark→bright). We evaluate four configurations: Baseline, Baseline+Prompt, Baseline+Memory, and the full VCoT. Adding motion prompts improves AUC from 0.539 to 0.542, showing that motion cues help compensate when appearance degrades. Adding memory further improves AUC to 0.561, indicating that accumulated historical cues remain reliable when current-frame appearance becomes unstable. When both are enabled, VCoT achieves the best AUC of 0.572, demonstrating their complementarity: motion prompts capture instantaneous dynamics, while memory provides cross-frame consistency, jointly improving robustness under illumination shifts.
> | Method                     | AUC   |
> |----------------------------|-------|
> | Baseline                   | 0.539 |
> | Baseline + Prompt          | 0.542 |
> | Baseline + Memory          | 0.561 |
> | Baseline + Prompt + Memory | 0.572 |
>
> (4)**Ensuring Fair Comparison with State-of-the-Art Trackers**
>
> For fair comparison, we re-trained the top three reproducible trackers (ODTrack, HIPTrack, AQATrack) using exactly the same day–night joint training data as our method. In the revised version, we additionally include the latest method H-DCPT, also re-trained under the same data setting, and report the full results. Please refer to Reviewer i22L for the experimental results.
>
> (5) **Clarification of the Three-Stage Reasoning Description**
> We agree with the reviewer that the previous description in the abstract may cause confusion. In the final version, we will consistently use the formulation “Observe–Recall&Infer–Memorize” to clearly denote the three-stage process and avoid ambiguity. We thank the reviewer for helping improve clarity.
>
> (6) **Regarding naming inconsistency**
>
>  We will unify all method names to ''VCoT (Ours)'' in the revised version.
>
> (7)**Complexity and Runtime Efficiency Analysis**
>
>  Following the reviewer’s suggestion, we have added efficiency statistics in the updated manuscript. VCoT runs at 30 FPS on a Titan X GPU, with 95M parameters and 23 MACs. These results demonstrate that VCoT satisfies real-time tracking requirements.
>
> | Tracker   | Params | MACs | Speed | Device  |
> |-----------|--------|------|-------|---------|
> | VCoT(Ours)   | 95M    | 23G  | 30fps | Titan X |
> | HIPTrack  | 120M   | 66G  | 25fps | Titan X |
> | ODTrack   | 92M    | 73G  | 16fps | Titan X |
>
> (8)**Memory Update Policy and Importance Estimation Strategy**
>
>  Memory writing and updating occur only during training. During testing, the memory state is frozen and used solely for prompt retrieval. We do not recompute importance during inference; instead, we use the gradient-based statistics obtained during training as fixed importance weights.

---

### Official Review · Reviewer_jKtE · 2025-10-30

**Soundness:** 3
**Presentation:** 3
**Contribution:** 3
**Rating:** 6
**Confidence:** 5

**Summary:**

This paper proposes a novel continuous learning framework for visual object tracking, VCoT, which achieves stable object tracking under both day and night lighting conditions.

The core idea is to introduce a human-like cognitive reasoning pipeline—Observe, Recall, Infer, and Memorize—to achieve structured, human-like adaptability. Experimental results on multiple benchmark datasets for day and night scenes demonstrate that VCoT maintains stable and superior performance under different lighting conditions, outperforming existing methods, including several strong baseline models.

**Strengths:**

1. The paper is good written

2. The concept in the field of tracking is very new.  Introducing a “Visual Chain-of-Thought” into object tracking sounds novel.

3. The paper achieved sota performance.

**Weaknesses:**

1.The comparison methods is insufficient. There are losts of sota Method need to be compared in 2025. There are also sota methods need to be compared, such as ARTrackV2, LoraT.

2.There are no efficiency comparisons and analysis (FPS, FLOPs), which are important for visual object tracking.

**Questions:**

Can authors provide more sota comparison and comlexity analysis?

---

> ### Author Response · Authors · 2025-11-20
>
> 1.**Response to comment**: (The comparison methods is insufficient. There are losts of sota Method need to be compared in 2025. There are also sota methods need to be compared, such as ARTrackV2, LoraT.)
>
> **Response**: We thank the reviewer for pointing out this limitation. Following your suggestion, we have added comparisons with several recent SOTA trackers to further validate the effectiveness of our method. Specifically, we include ARTrack, MCITrack, and H-DCPT in the revised manuscript (page 7, Table 2 and Table 3). In addition, on the LaSOT, TrackingNet, and GOT-10k datasets, we also report results for ARTrackV2 and LoraT (page 8, Table 4, Table 5, Table 6). These new experiments consistently show that VCoT maintains strong performance across diverse benchmarks. Please refer to Reviewer i22L for the experimental results.
>
> 2.**Response to comment**: (There are no efficiency comparisons and analysis (FPS, FLOPs), which are important for visual object tracking.)
>
> **Response**: We agree with the reviewer that efficiency is crucial for visual object tracking. Therefore, we have added detailed efficiency metrics in the revised version (page 7, Table 1). VCoT contains 95M parameters and requires 23 MACs for a single forward pass. In our actual deployment tests on a Titan X GPU, VCoT achieves 30 FPS, meeting the real-time requirement of  tracking.
>
> | Tracker   | Params | MACs | Speed | Device  |
> |-----------|--------|------|-------|---------|
> | VCoT(Ours)   | 95M    | 23G  | 30fps | Titan X |
> | HIPTrack  | 120M   | 66G  | 25fps | Titan X |
> | ODTrack   | 92M    | 73G  | 16fps | Titan X |

---

### Official Review · Reviewer_i22L · 2025-11-01

**Soundness:** 2
**Presentation:** 2
**Contribution:** 2
**Rating:** 4
**Confidence:** 5

**Summary:**

The authors tackle the weaknesses of conventional visual tracking algorithms, where they solely rely on the appearance feature representations. This paper proposes to employ visual chain-of-thought (VCoT) for visual tracking formulation, where the authors formulate this as observe-recall-infer-memorize steps. The authors claim that they are the first to employ CoT formulation for visual tracking task, and the experimental results show strong results on multiple benchmark datasets, especially for tracking under low-light night conditions.

**Strengths:**

- The authors propose an interesting approach for solving the visual tracking task, where they redefine the visual tracking process apart from conventional tracking algorithms.

- The objective of the proposed method is explained in a straightforward manner, with simple formulation which can be easily implemented and reproduced by other researchers.

- The authors performed extensive experimental validations and comparisons on multiple well-known visual tracking datasets and algorithms, and they also conducted some ablation experiments to facilitate further understanding for the proposed work.

**Weaknesses:**

- The chain-of-thought (CoT) concept used in this paper seems to be largely deviated from the original concept used in the natural language processing (NLP) field. Generally, CoT is thought to be a dynamic reasoning process that is variable in length, and its steps include sequential causal chains that are interpretable and flexible enough for reaching diverse conclusions at the end. However, the so-called CoT in this paper seems to be a fixed pipeline, with concrete heuristic steps of observe-recall-infer-memorize, with no room for logical reasoning and sequential process.

- The proposed method explicitly uses nighttime datasets (BDD100K-Night, SHIFT-Night) in a continual learning setup, whereas most comparison baselines are trained only on daytime datasets. Fair comparison with equivalent training datasets and backbone network should be conducted.

- Is the "continual learning" formulation necessary? The proposed gradient-based memory retention essentially acts as a simple replay buffer rather than a principled continual learning algorithm, and no comparison to well-known continual learning baselines such as EwC and LwF are not conducted.

**Questions:**

Please refer to the weaknesses section. Although the authors' take on the visual tracking task is appealing, the machine learning concepts used in the proposed method seem to be overstated and deviated from its original form.

---

> ### Author Response · Authors · 2025-11-20
>
> 1.**Response to comment**: (The chain-of-thought (CoT) concept used in this paper seems to be largely deviated from the original concept used in the natural language processing (NLP) field. Generally, CoT is thought to be a dynamic reasoning process that is variable in length, and its steps include sequential causal chains that are interpretable and flexible enough for reaching diverse conclusions at the end. However, the so-called CoT in this paper seems to be a fixed pipeline, with concrete heuristic steps of observe-recall-infer-memorize, with no room for logical reasoning and sequential process.)
>
> **Response**: Thank you for the reviewer’s comments. The VCoT used in this paper is a simplified form of CoT tailored for visual tracking, rather than an attempt to reproduce the variable-length, dynamically generated reasoning chains used in NLP. In NLP, CoT can extend its reasoning steps through symbolic sequences, whereas efficient single-object tracking requires frame-by-frame outputs with lightweight, stable, and computationally controllable inference. A dynamic CoT would significantly increase parameters and inference latency, and the varying reasoning length across frames could introduce instability and tracking drift. Considering these task constraints, we retain the core CoT idea of “multi-stage reasoning with shared contextual information across stages,” and implement it as the fixed and efficient Observe–Recall&Infer–Memorize process. This design provides the benefits of structured reasoning while ensuring  efficiency and model stability. We appreciate the reviewer’s expectation for a more NLP-style dynamic CoT, and this is indeed an important direction for future work.  We plan to explore more flexible visual CoT structures in future research to further bridge the gap toward NLP-style CoT. The framework presented in this paper can be viewed as an initial and feasible step toward visual CoT, laying the groundwork for more sophisticated and dynamic reasoning mechanisms.
>
>
>
> 2.**Response to comment**: (The proposed method explicitly uses nighttime datasets (BDD100K-Night, SHIFT-Night) in a continual learning setup, whereas most comparison baselines are trained only on daytime datasets. Fair comparison with equivalent training datasets and backbone network should be conducted.)
>
> **Response**: Thank you for the reviewer’s valuable comments. We understand your concerns regarding experimental fairness. To ensure fair comparisons, we have retrained three high-performing methods for which public training code is available—ODTrack, HIPTrack, and AQATrack—using the same day–night training dataset as VCoT. In the revised manuscript, we have also added the latest method, H-DCPT, which has likewise been retrained and evaluated on the same dataset to further guarantee fairness and comparability.
>
>
>
> 3.**Response to comment**: (Is the "continual learning" formulation necessary? The proposed gradient-based memory retention essentially acts as a simple replay buffer rather than a principled continual learning algorithm, and no comparison to well-known continual learning baselines such as EwC and LwF are not conducted.)
>
> **Response**: Thank you for the reviewer’s comments. From a definitional perspective, continual learning does not require the use of complex parameter regularization, dynamic subnetwork expansion, or generative replay; memory replay or experience retention is itself one of the most classical and widely accepted CL paradigms. Therefore, the reviewer’s observation that our method resembles a replay buffer is not in conflict with the fundamental concept of continual learning—replay-based strategies are in fact among the most recognized and stable CL approaches.
>
> Second, our mechanism is not a traditional replay buffer, as we do not store or replay raw training samples. Instead, we retain task-relevant intermediate representations in the form of prompt vectors and evaluate their importance through gradient sensitivity. This design aligns with the emerging prompt-based continual learning (Prompt-CL) paradigm in recent CL researches. Our method follows the core idea of Prompt-CL: using prompts as a stable interface for cross-task knowledge transfer, and mitigating forgetting through selective preservation and fusion of prompts. Thus, unlike sample-level replay, our method represents a natural extension of Prompt-CL, focusing on maintaining crucial prompt representations rather than replaying historical data, thereby achieving cross-task knowledge retention in a lightweight and structured manner.
>
> We fully understand the reviewer’s interest in comparing against additional CL baselines. Since no prior work has introduced a continual learning framework into single-object tracking, there are no  CL-for-tracking baselines that can be directly reproduced.  During the rebuttal period we have therefore added the most feasible classical CL baseline—replay-based CL.

---

> ### Author Response · Authors · 2025-11-20
>
> | Method          | Source   |    \|    | UAVDark135 |     \|     |    \|    | NAT2024-1 |     \|     |    \|    | DarkTrack2021 |     \|     |
> | --------------- | -------- | :------: | :--------: | :--------: | :------: | :-------: | :--------: | :------: | :-----------: | :--------: |
> |                 |          | **AUC**  |   **P**    | **P_Norm** |  **AUC**  |   **P**   | **P_Norm** | **AUC**  |     **P**     | **P_Norm** |
> | **VCoT (Ours)** | -        | **68.6** |  **82.6**  |  **83.9**  | **73.3** | **94.5**  |  **90.7**  | **62.6** |     74.2      |    75.0    |
> | MambaLCT        | AAAI2025 |   64.4   |    77.9    |    80.6    |   60.8   |   88.9    |    85.8    |   61.5   |     74.4      |  **75.1**  |
> | H-DCPT          | T-IV2025 |   64.7   |    77.3    |    77.9    |   71.0   |   90.8    |    86.2    |   62.0   |   **74.6**    |    74.2    |
> | ARTrack         | CVPR2023 |   65.5   |    77.8    |    79.3    |   69.9   |   90.9    |    84.3    |   61.2   |     72.5      |    72.0    |
> | ODTrack         | AAAI2024 |   63.2   |    77.8    |    78.1    |   69.1   |   89.6    |    85.5    |   60.5   |     72.2      |    72.3    |
> | MCITrack        | AAAI2025 |   58.4   |    68.7    |    70.3    |   65.2   |   81.8    |    78.2    |   54.5   |     64.8      |    65.4    |
> | HIPTrack        | CVPR2024 |   59.7   |    72.0    |    70.0    |   69.4   |   88.4    |    84.1    |   57.1   |     68.5      |    68.7    |
> | EVPTtrack       | AAAI2024 |   58.1   |    69.2    |    70.5    |   65.0   |   83.7    |    78.1    |   53.7   |     64.8      |    64.7    |
> | AQATrack        | CVPR2024 |   58.2   |    69.2    |    70.7    |   64.2   |   82.1    |    77.1    |   55.0   |     66.1      |    66.8    |
> | SAM-DA          | ICARM24  |   47.6   |    60.4    |    59.4    |   53.4   |   75.3    |    64.9    |   44.7   |     55.5      |    54.6    |
> | DCPT            | ICRA2024 |   56.7   |    69.2    |    69.8    |   62.1   |   80.9    |    75.4    |   54.0   |     66.7      |    64.6    |
> | AVTrack         | ICML2024 |   47.6   |    58.6    |    59.2    |   56.7   |   68.2    |    75.3    |   46.1   |     55.1      |    54.9    |
> | LiteTrack       | CVPR2024 |   53.9   |    63.6    |    65.9    |   61.8   |   79.7    |    74.1    |   52.8   |     63.5      |    62.8    |
>
>
>
>
>
>
> | Method          | Source   |    \|    |  DTB70   |     \|     |    \|    | VisDrone2018 |     \|     |    \|    |  UAVDT   |     \|     |
> | --------------- | -------- | :------: | :------: | :--------: | :------: | :----------: | :--------: | :------: | :------: | :--------: |
> |                 |       | **AUC**  |  **P**   | **P_Norm** | **AUC**  |    **P**     | **P_Norm** | **AUC**  |  **P**   | **P_Norm** |
> | **VCoT (Ours)** | -        | **70.1** | **90.7** |    85.3    | **70.6** |   **90.0**   |    86.8    | **67.2** | **88.8** |  **77.6**  |
> | MambaLCT        | AAAI2025 |   68.7   |   88.3   |    84.0    |   65.4   |     88.1     |    84.3    |   63.6   |   84.4   |    75.8    |
> | H-DCPT          | T-IV2025 |   68.6   |   77.3   |    77.9    |   68.5   |     88.5     |    85.6    |   61.8   |   81.6   |    74.1    |
> | ARTrack         | CVPR2023 |   66.4   |   87.5   |    81.1    |   65.9   |     87.4     |    82.9    |   63.8   |   85.1   |    74.0    |
> | ODTrack         | AAAI2024 |   70.0   |   90.0   |  **86.1**  |   64.7   |     85.6     |    83.1    |   63.8   |   85.8   |    74.7    |
> | MCITrack        | AAAI2025 |   68.1   |   87.4   |    82.0    |   66.7   |     84.3     |    81.2    |   62.1   |   81.5   |    73.3    |
> | HIPTrack        | CVPR2024 |   68.6   |   81.2   |    76.2    |   67.1   |     86.7     |    83.9    |   60.9   |   81.2   |    76.2    |
> | EVPTtrack       | AAAI2024 |   66.6   |   86.7   |    81.7    |   66.6   |     87.0     |    82.6    |   60.2   |   80.0   |    71.3    |
> | AQATrack        | CVPR2024 |   66.1   |   86.3   |    80.7    |   66.9   |     87.2     |  **89.8**  |   63.7   |   84.7   |    75.9    |
> | SAM-DA          | ICARM24  |   63.0   |   82.2   |    76.3    |   53.1   |     71.4     |    67.0    |   61.3   |   82.6   |    73.3    |
> | DCPT            | ICRA2024 |   64.6   |   83.7   |    77.6    |   64.2   |     83.1     |    79.7    |   56.9   |   76.8   |    66.0    |
> | AVTrack         | ICML2024 |   65.0   |   84.3   |    80.0    |   64.2   |     84.8     |    80.3    |   58.7   |   82.1   |    68.6    |
> | LiteTrack       | CVPR2024 |   64.7   |   83.5   |    77.6    |   61.8   |     79.8     |    75.7    |   62.1   |   84.3   |    71.6    |
>
>
>
> | Train Data         | Method   | Succ. | Prec. |
> |--------------------|----------|-------|-------|
> | Daytime only       | Baseline | 70.99 | 77.04 |
> | Daytime + Nighttime| Baseline | 67.49 | 72.71 |
> | Daytime + Nighttime| Replay   | 69.45 | 75.13 |
> | Daytime + Nighttime| VCoT     | 72.74 | 78.87 |

---

### Meta-Review · Area_Chair_xJCs · 2026-01-08

**Summary:**

This paper proposes VCoT, a visual tracking framework that reformulates object tracking as a structured reasoning process of Observe-Recall-Infer-Memorize to handle drastic illumination changes. It utilizes a gradient-based importance evaluation to selectively retain historical prompts in a memory pool, enabling continual learning and knowledge transfer between daytime and nighttime scenarios.
1. Reviewers i22L and 5Xte pointed out that the "Chain-of-Thought" (CoT) concept in this paper deviates from the original NLP definition, appearing more like a fixed heuristic pipeline rather than a dynamic reasoning process.
2. Reviewer i22L and RZyp noted that the experimental comparison was unfair or incomplete, as baselines were often trained only on daytime data while VCoT used nighttime datasets, and several recent SOTA trackers were missing.
3. Reviewer i22L questioned the necessity of the "continual learning" formulation, arguing that the gradient-based memory acts merely as a simple replay buffer without comparison to principled CL baselines like EwC or LwF.
4.  Reviewer jKtE and 5Xte expressed concerns regarding the lack of efficiency analysis (FPS, FLOPs) and missing implementation details (e.g., prompt length, memory capacity), which limit reproducibility.
5. Reviewer RZyp argued that the paper lacks novelty, suggesting the method is fundamentally conventional temporal processing and overlaps significantly with existing works like ARTrack and MixViT.

Considering the overstated conceptual framing of "Chain-of-Thought" and the incremental technical novelty over existing memory-based trackers, I tend to recommend rejection.

**Reviewer Concerns:**

1. The authors clarified that VCoT is a simplified form tailored for tracking stability and efficiency; however, I believe the gap between this fixed pipeline and true "reasoning" remains significant, making the CoT branding feel somewhat overstated.
2.. The authors retrained several high-performing methods (ODTrack, HIPTrack, etc.) on the same day-night datasets and added comparisons with ARTrackV2 and LoraT; I believe this effectively addresses the fairness and SOTA comparison concerns.
3. The authors argued that prompt-based memory retention aligns with the Prompt-CL paradigm and added a replay-based CL baseline; however, the distinction from a standard replay buffer remains thin, and the "continual learning" framing still feels forced for this specific task.
4. The authors provided detailed efficiency metrics (30 FPS, 95M params) and clarified hyper-parameters ($L=5$, $M=50$); I believe this successfully resolves the concerns regarding transparency and reproducibility.
5. The authors highlighted the differences in their structured reasoning loop compared to autoregressive or simple fusion models; however, the technical components (attention over memory, gradient-based selection) are indeed quite standard in recent literature, leaving the novelty claim weak.

**Reviewer Scores:**

- Reviewer i22L: 4 -> 4, although the authors added experiments, the deviation of the CoT concept and the simplistic "continual learning" formulation remain fundamental conceptual weaknesses.

- Reviewer jKtE: 6 -> 8, the concerns regarding SOTA comparisons and efficiency analysis were well-addressed with comprehensive new data and tables.

- Reviewer 5Xte: 6 -> 6, while implementation details were clarified, the reviewer's core doubt about the incremental nature of the "reasoning" capability beyond existing memory-augmented trackers persists.

- Reviewer RZyp: 2 -> 2, the author's response failed to convincingly differentiate the method from existing temporal trackers, and the concern regarding limited novelty and restricted application scenarios remains unresolved.

---

### Decision · Program_Chairs · 2026-01-26

Reject